# *Gnas* Promoter Hypermethylation in the Basolateral Amygdala Regulates Reconsolidation of Morphine Reward Memory in Rats

**DOI:** 10.3390/genes13030553

**Published:** 2022-03-21

**Authors:** Peng Liu, Jialong Liang, Fengze Jiang, Wanshi Cai, Fang Shen, Jing Liang, Jianjun Zhang, Zhongsheng Sun, Nan Sui

**Affiliations:** 1CAS Key Laboratory of Mental Health, Institute of Psychology, Beijing 100101, China; shk1985@126.com (P.L.); gracejiang007@163.com (F.J.); shenf@psych.ac.cn (F.S.); liangj@psych.ac.cn (J.L.); suin@psych.ac.cn (N.S.); 2Department of Psychology, University of Chinese Academy of Sciences, Beijing 100101, China; 3Beijing Institutes of Life Science, Chinese Academy of Sciences, Beijing 100101, China; lljjll.cool@163.com (J.L.); caiwanshi@126.com (W.C.); 4Institute of Genomic Medicine, Wenzhou Medical University, Wenzhou 325000, China

**Keywords:** DNA methylation, DNA methyltransferases, morphine, basolateral amygdala, reconsolidation, memory

## Abstract

Impairing reconsolidation may disrupt drug memories to prevent relapse, meanwhile long-term transcription regulations in the brain regions contribute to the occurrence of emotional memories. The basolateral amygdala (BLA) is involved in the drug-cue association, while the nucleus accumbens (NAc) responds to the drug reward. Here, we assessed whether DNA methyltransferases (Dnmts) in these two brain regions function identically in the reconsolidation of morphine reward memory. We show that Dnmts inhibition in the BLA but not in the NAc after memory retrieval impaired reconsolidation of a morphine reward memory. Moreover, the mRNA levels of *Dnmt3a* and *Dnmt3b*, rather than *Dnmt1*, in the BLA were continuously upregulated after retrieval. We further identified the differentially methylated regions (DMRs) in genes in the BLA after retrieval, and focused on the DMRs located in gene promoter regions. Among them were three genes (*Gnas*, *Sox10*, and *Pik3r1*) involved in memory modulation. Furthermore, *Gnas* promoter hypermethylation was confirmed to be inversely correlated with the downregulation of *Gnas* mRNA levels. The findings indicate that the specific transcription regulation mechanism in the BLA and NAc on reconsolidation of opiate-associated memories can be dissociable, and DNA hypermethylation of *Gnas* in the BLA is necessary for the reconsolidation of morphine reward memories.

## 1. Introduction

After retrieval, acquired drug memories are transferred to a labile process termed reconsolidation. Numerous human and animal investigations have shown that disrupting reconsolidation offers the first realistic opportunity to weaken drug memories, thereby reducing drug craving and drug-seeking behavior [1,2,3,4]. The blockade of time-limited reconsolidation [5,6] persistently disrupts memory in experimental animals [6,7,8]. In general, those studies have focused on critical events (such as receptor activation, protein kinase activation, and protein synthesis) in reconsolidation, but little is known about how these events lead to the maintenance of impaired memory. One possibility is that transcription regulation occurs during reconsolidation and results in the maintenance of altered transcriptional status. Epigenetics, especially DNA methylation catalyzed by DNA methyltransferases (Dnmts) [9], may be a particularly promising candidate mechanism by which potentially stable and long-lasting effects on gene expression are exerted [10,11].

Recent studies suggest that the intervention of reconsolidation does not make one forget what happened, rather it makes one less emotionally affected when remembering what happened as if the emotional edge were removed from the memory [12,13]. Although the basolateral amygdala (BLA) and nucleus accumbens (NAc) are two of the most important brain regions in the reconsolidation of drug-related memories [7], the BLA but not the NAc may be responsible for the removal of the emotional edge during the reconsolidation, because the BLA integrates the drug-cue association with affect value. The NAc, on the other hand, may only amplify this association [14]. This may be one reason why extracellular signal-regulated kinase in the BLA but not in the NAc is involved in the reconsolidation [15]. Thus, we also suspect that DNA methylation is critical for reconsolidation through the silencing and stabilization of gene expression in the BLA but not in the NAc.

Despite the important roles DNA methylation play in drug addiction, the evidence that DNA methylation is critical for reconsolidation of drug memory is lacking. Previous studies reported that DNA methylation contributed to cocaine-induced spine plasticity [16,17] and was associated with memory modification processes by means of persistent transcriptional repression [16,17,18,19,20]. We also found that DNA methylation was region-specifically involved in the acquisition, consolidation, and retrieval of drug reward memory [21,22,23].

In the present study, using a rat model of morphine-induced conditioned place preference (mCPP), we examined the effect of Dnmt inhibition in the amygdala (BLA and central amygdala, CeA) and NAc (core and shell) on the reconsolidation. We also explored the subtypes of Dnmts, the transcriptions of which were regulated in the reconsolidation. Finally, we investigated the key target genes modified by DNA methylation during the reconsolidation by function.

## 2. Materials and Methods

### 2.1. Subjects

Male Sprague–Dawley rats, weighing 220–250 g before surgery, were individually housed in metal mesh cages (25 cm × 22.5 cm × 30 cm) under a 12-h:12-h light/dark cycle (lights on at 08:00) with food and water available *ad libitum*. The experiments were performed in accordance with the National Institute of Health Guide for the Care and Use of Laboratory Animals. All the protocols were approved by the Research Ethics Review Board of Institute of Psychology, Chinese Academy of Sciences.

### 2.2. Drugs

Morphine hydrochloride (Qinghai Pharmaceutical, Xining, China, H63020013) was dissolved in sterilized physiological saline to a concentration of 5 mg/mL and administered intraperitoneally (i.p.) at 1.0 mL/kg body weight. The DNA methyltransferase inhibitor 5-aza-2′-deoxycytidine (5-aza, Sigma–Aldrich, St. Louis, MO, USA, 2353-33-5) was dissolved in 0.8% acetic acid (AcOH) at a concentration of 2 µg/µL or 0.2 µg/µL [22,24].

### 2.3. Experimental Procedures

Intracranial surgery: Rats were anesthetized with sodium pentobarbital (75 mg/kg, i.p.) and mounted in a stereotaxic frame (Stoelting Co., Dale, IL, USA). Stainless steel guide cannulas (o.d. 0.6 mm, i.d. 0.35 mm, length 11.5 mm for the BLA, 11 mm for the CeA, 9 mm for the NAc shell and NAc core) were bilaterally implanted 1 mm above the target sites, according to the following coordinates [25]: BLA (−2.8 mm A/P; ±4.9 mm M/L; −8.5 mm D/V), CeA (−2.6 mm; ±4.2 mm M/L; −7.9 mm D/V), NAc shell (+1.6 mm A/P; ±0.8 mm M/L; −7.0 mm D/V), NAc core (+1.6 mm A/P; ±1.6 mm M/L, −6.8 mm D/V). Cannulas were anchored to the skull by three small skull screws and dental cement. Stylets were inserted into each cannula to prevent occlusion. All rats were injected with penicillin (80,000 units, i.p.) to prevent infection and allowed seven days to recover from cannula insertion before conducting behavioral tests.

Intracranial injections: The stylets were removed, and infusion cannulas were inserted. The injectors extended 1 mm below the tips of the guide cannulas. The cannulas were connected to a 10-µL Hamilton microsyringe via polyethylene tubing. Drugs or solvents were bilaterally injected (0.5 µL/side) at a rate of 0.25 µL/min, and the injectors were left in place for an additional 2 min to prevent backflow.

Apparatus: The apparatus was a polyvinyl chloride box containing three compartments separated by guillotine doors. The two larger compartments (35 cm × 31 cm × 40 cm) were separated by a smaller compartment (14 cm × 31 cm × 40 cm). One of the larger compartments had a grid-textured floor and horizontal stripes on the wall, and the other had a bar-textured floor and vertical stripes on the wall. The location of the rats was monitored by a video camera suspended above the apparatus. The video data were analyzed using professional software (Shanghaiyishu Company, Shanghai, China).

#### Conditioned Place Preference

Pre-Conditioning (Pre-C) test. On day 1, rats were allowed to explore all three compartments freely for 15 min to assess baseline preferences. The rats that showed a strong initial preference (the time that rats spent in any compartment > 540 s) were excluded.

Conditioning. On days 2, 4, 6, and 8, each rat was confined to its saline-paired compartment for 45 min after injection of saline (1 mL/kg, i.p.). On days 3, 5, 7, and 9, the rat was confined to the morphine-paired compartment for 45 min after injection of morphine (5 mg/kg, i.p.). The dose of morphine was selected based on our previous work [26].

Post-conditioning (Post-C) test. On day 11, rats were allowed to explore all three compartments freely for 15 min. The CPP score was calculated as the time that rats spent in the morphine-paired side divided by the total time spent in both the morphine- and saline-paired sides.

Memory retrieval and retention test. On day 12, rats were allowed to explore all compartments freely for 10 min [3], as memory retrieval, to induce reconsolidation. The preference scores were assessed on days 13, 19, and 26 (retention test: RT1, RT7, and RT14, respectively; 15 min each).

Histological verification of infusion sites: Rats were deeply anesthetized with chloral hydrate and perfused transcardially with physiological saline, followed by 4% paraformaldehyde (PFA) in 0.1 M phosphate-buffered saline (PBS, pH 7.4). We extracted the brains and postfixed them in 4% PFA overnight and then transferred them to a 30% sucrose in PBS solution for four days at 4 °C. A freezing microtome was used to obtain coronal sections (40-µm-thick). Sections were stained with Nissl staining, and the placement of the cannulas was verified under a light microscope (Appendix A). Seventy-eight rats were excluded due to incorrect cannula placements.

Isolation of the nucleus: Rats were sacrificed by rapid decapitation at different time points after the memory retrieval. Brains were immediately removed and flash-frozen on dry ice and then stored at −80 °C. Nucleus sections were removed with a 1-mm punch tool in a freezing microtome and processed for RT-qPCR or Reduced-representation bisulphite sequencing (RRBS).

*RT-qPCR:* BLA punches were processed for mRNA quantification. Total RNA was extracted using the RNeasy Mini kit (Qiagen, Hilden, Germany, 74104) following the manufacturer’s instructions. mRNA was reverse transcribed using the iScript RT-PCR kit (Bio-Rad, CA, USA, 1708895). Specific Intron-spanning primers were used to amplify cDNA regions for transcripts of interest (Dnmt1, Dnmt3a, Dnmt3b, and Gnas; see Table 1 for primer sequences). qPCR amplifications were performed in triplicate using a real-time PCR system (Eppendorf, Hamburg, Germany) at 95 °C for 10 min, followed by 40 cycles each consisting of 95 °C for 30 s, and then incubation at 72 °C for 10 min, followed by real-time melt analysis to verify product specificity. Gapdh was used as an internal control for level normalization using the ΔCt method.

RRBS library preparation and sequencing: DNA was extracted from the tissue by using Qiagen DNeasy Blood & Tissue kit (Qiagen, Hilden, Germany, 69504) and quantified using a Qubit 2.0 fluorometer with the Qubit dsDNA H.S. Assay kit (Invitrogen, Carlsbad, CA, USA, Q32854). To prepare the RRBS library, 200 ng of DNA was digested with MspI restriction endonuclease (New England Biolabs, Ipswich, MA, USA, R0106L). The MspI-digested DNA fragments were then subjected to end-repairing and A-tailing using Klenow fragment (3′→5′ exo-) (New England Biolabs, M0212L). The A-tailed DNA was further ligated to Illumina adapters with all Cs methylated. The ligated DNA was bisulphite-converted and purified using an EZ DNA Methylation Gold^TM^ kit (Zymo Research, Irvine, CA, USA, D5005) according to the manufacturer’s standard protocol. The bisulphite-converted DNA was then amplified by PCR for 16 cycles using two × Quest Taq Premix (Zymo Research, Irvine, CA, USA, E2051). Size selection of a 160−340-bp library was performed in a 2% agarose gel, and DNA was recovered from the gel by using a Qiagen MinElute Gel Extraction kit (Qiagen, 28604). The final library was quantified using a Qubit 2.0 Fluorometer with a Qubit dsDNA HS Assay kit (Invitrogen, Q32854) and sequenced on an Illumina HiSeq 2000 analyzer (Illumina, San Diego, CA, USA, SY-401-1001) to generate paired-end reads of 150 bp.

Identification of differentially methylated regions (DMRs): High-quality clean reads in RRBS data were generated by filtering out low-quality reads and cutting adaptor sequences. The remaining truncated reads were then aligned to the rn5 rat reference genome (downloaded from UCSC Genome Browser) with the Bismark tool, using the default parameters [27]. Uniquely aligned reads that contained MspI digestion sites at their ends were retained for further analyses. The bisulphite conversion rate was determined by calculating the frequency of unconverted cytosines in non-CpG regions. The position of methylcytosines and the methylation pattern of each methylcytosine were then determined by using Bismark. DMRs were then identified by swDMR (fold change > 1.25, *p* < 0.05, CpG number ≥ 5) [28].

Validation of DMRs: DNA was bisulphite-converted and purified by using an EZ DNA Methylation Gold^TM^ kit (Zymo Research, D5005) according to the manufacturer’s standard protocol. Nested PCR primers for amplifying target regions were designed using Methprimer (http://www.urogene.org/methprimer/, accessed on 13 December 2016). Nested PCR was performed using Quest Taq Premix (Zymo Research, E2051). The thermal cycle profile for PCR amplifying bisulphite-treated DNA was as follows: 95 °C for 10 min; 40 cycles consisting of 95 °C for 30 s, 50 °C for 30 s, and 72 °C for 45 s; and finally, 72 °C for 10 min. The PCR products of the same sample were pooled together and used for preparing Illumina libraries. The libraries were sequenced on an Illumina HiSeq 2000 analyzer (Illumina, San Diego, CA, USA, SY-401-1001) to generate paired-end reads of 150 bp.

### 2.4. Experimental Design and Statistical Analysis

**Experiment** **1.**
*We set out to investigate whether the inhibition of DNMT in the BLA or the NAc could impair the reconsolidation of morphine reward memory. Immediately after retrieval, 0.8% acetic acid (AcOH) or the Dnmts inhibitor 5-aza was microinjected into the BLA, central amygdala (CeA), NAc shell, or NAc core (n = 7–11). Previously, we injected 5-aza into the BLA and did not find that it affected the locomotor activities or the preference/aversion of drug-naive rats [24]. The preference scores were assessed on retention tests one, seven, and 14 days after retrieval.*


**Experiment** **2.**
*The goal of Experiment 2 was to identify whether the effects of 5-aza on the retention tests were due to the impairment of memory reconsolidation. Groups of rats were microinjected into the BLA with either AcOH (n = 13) or 5-aza (n = 13) on the day after the post-conditioning (Post-C) test, without memory retrieval. Separate groups of rats were administered BLA microinjections 1 h (AcOH, n = 12; 5-aza, n = 9) or 12 h (AcOH, n = 8; 5-aza, n = 7) after memory retrieval. The preference scores were assessed on retention tests one, seven, and 14 days after retrieval.*


**Experiment** **3.**
*Here we tested which subtypes of Dnmt and when the Dnmt were upregulated in the BLA during the reconsolidation. At 0 h, 0.5 h, 1 h, 3 h, 6 h, and 12 h after memory retrieval, rats were sacrificed, the brains quickly removed, and the BLA were punched out and used for gene expression analysis by RT-PCR (n = 8–10).*


**Experiment** **4.**
*We then tested the key genes that are methylated during memory reconsolidation. DNA methylation was analyzed 1 h after memory retrieval. RRBS analysis was performed to investigate the DNA methylation level in the BLA of the two groups of rats: retrieval vs. no-retrieval groups (two samples each, each sample consisted of the pooled BLA tissues of six rats). The DMRs were identified (difference in DNA methylation level > 0.2, p < 0.05, CpG number ≥ 5). Using bisulphate PCR, we further validated the DMRs located in the promoter regions.*


### 2.5. Statistical Analysis

Data are presented as mean ± SEM. We analyzed the behavioral data using two-way repeated-measures analysis of variance (ANOVA, treatment: 5-aza/AcOH or overexpression virus/control virus as a between-subject factor; test point: Pre-C, Post-C, RT1, RT7, and RT14 as within-subject factor.) or *t*-test. We analyzed the molecular data using two-way ANOVA (treatment as a between-subject factor: retrieval/no retrieval; time-points as a within-subject factor: 0 h, 0.5 h, 1 h, 3 h, 6 h, 12 h). For significant interactions or main effects, we performed Bonferroni *post hoc* tests. Differences in mRNA expression were compared with an unpaired Student’s *t*-test. Statistical significance was designated at *p* = 0.05 for all analyses. Statistical analyses were performed with GraphPad (Prism 6, GraphPad, La Jolla, CA, USA).

## 3. Results

### 3.1. Region-Specific Effects of Dnmts Inhibition on the Reconsolidation of Morphine Reward Memory

To examine the potential role of DNA methylation in the reconsolidation of morphine reward memory, we investigated whether Dnmts activity was required for reconsolidation. Immediately after a 10-min memory retrieval, 0.8% acetic acid (AcOH) or the Dnmts inhibitor 5-aza was microinjected into the BLA (AcOH: *n* = 10; 0.2 µg 5-aza: *n* = 10; 2 µg 5-aza: *n* = 8), CeA (AcOH: *n* = 10; 0.2 µg 5-aza: *n* = 9; 2 µg 5-aza: *n* = 9), NAc shell (AcOH: *n* = 7; 2 µg 5-aza: *n* = 7), or NAc core (AcOH: *n* = 11; 2 µg 5-aza: *n* = 7) (Appendix A). In rats that received BLA treatment, ANOVA showed significant effects of test point (*F*_4, 100_ = 14.89, *p* < 0.001) and interaction between test point and treatment (*F*_8, 100_ = 2.16, *p* < 0.05), but the effect of treatment was not significant (*F*_2, 25_ = 1.79, *p* = 0.19). *Post hoc* analyses revealed that 2 µg 5-aza injections into the BLA decreased the mCPP score at the retention test (RT) 14 (*p* < 0.01, Figure 1b). Microinjection of 5-aza into the CeA, NAc shell, or NAc core had no effect on the RTs (all *p* > 0.05, Figure 1c–e). In summary, the injection of 5-aza into the BLA, but not the CeA or NAc, impaired reconsolidation of the established mCPP, suggesting that Dnmts in the BLA play a key role in reconsolidation.

According to previous evidence, reconsolidation could be retrieval-dependent, and it could be hindered by pharmacological or non-pharmacological manipulations within 6 h after memory retrieval [3]. Thus, we examined whether the effect of 5-aza on the mCPP was retrieval-dependent. Rats were given microinfusions of AcOH (*n* = 13) or 5-aza (*n* = 13) into the BLA on the day after the post-conditioning (Post-C) test, without memory retrieval, which was followed by microinfusion-free CPP tests once a week (RT1, RT7, RT14; Figure 2b). ANOVA revealed no significant effects of treatment (*F*_1, 24_ = 0.07, *p* > 0.05), test point (*F*_3,_
_72_ = 0.38, *p* > 0.05), or interaction between these two factors (*F*_3,_
_72_ = 0.93, *p* > 0.05).

We also identified the temporal window for this effect of intra-BLA Dnmts blockade on reconsolidation of the morphine reward memory. We selected the 1-h and 12-h time-points as being within and outside of the reconsolidation window, respectively, to verify this. AcOH (*n* = 12) or 5-aza (*n* = 9) was infused into the BLA 1 h after memory retrieval (Figure 2c), and ANOVA revealed significant effects of treatment (*F*_1, 19_ = 23.98, *p* < 0.001), test point (*F*_4, 76_ = 7.15, *p* < 0.001), and their interaction (*F*_4, 76_ = 4.41, *p* < 0.05). The *post hoc* analyses showed that 5-aza-treated rats spent less time in the chamber previously paired with morphine than did control group rats at RT1 (*p* < 0.01), RT7 (*p* < 0.001), and RT14 (*p* < 0.001) test points. Moreover, morphine priming failed to reinstate the mCPP on day 27 (*t* = 0.08, *p* = 0.94, Reinstatement vs. RT14).

A separate group of rats was administered BLA microinjections of AcOH (*n* = 8) or 5-aza (*n* = 7) at 12 h after memory retrieval (Figure 2d). ANOVA showed a significant effect of test point (*F*_4,_
_52_ = 9.20, *p* < 0.001), but no significant effect of treatment (*F*_1, 13_ = 0.05, *p* = 0.82) or their interaction (*F*_4,_
_52_ = 1.37, *p* = 0.26).

These data confirmed that Dnmts inhibition in the BLA selectively prevented the reconsolidation of mCPP.

### 3.2. Upregulation of Dnmt3a and Dnmt3b mRNA Expression in the BLA after Memory Retrieval

To investigate when Dnmts are upregulated during the reconsolidation phase, we performed RT-PCR to evaluate the transcription of Dnmts (*Dnmt1*, *Dnmt3a*, and *Dnmt3b*) in the BLA at 0 h, 0.5 h, 1 h, 3 h, 6 h, and 12 h after memory retrieval (Figure 3a, *n* = 8–10). For *Dnmt3a* and *Dnmt3b*, ANOVA revealed a significant main effect of retrieval condition (retrieval vs. no retrieval) (*Dnmt3a*: *F*_1, 103_ = 36.40, *p* < 0.001; *Dnmt3b*: *F*_1, 97_ = 13.76, *p* < 0.001). There was no significant main effect of test or interaction (Dnmt3a: *F*_5, 103, test_ = 2.00, *p* = 0.08; *F*_5, 103, interaction_ = 0.82, *p* = 0.54; Dnmt3b: *F*_5, 97, test_ = 1.87, *p* = 0.11; *F*_5, 97, interaction_ = 1.78, *p* = 0.12). *Post hoc* analyses showed that the mRNA levels of *Dnmt3a* were significantly increased at 1 h (*p* < 0.01) and 12 h (*p* < 0.01) after memory retrieval. Similarly, the mRNA level of *Dnmt3b* was significantly increased at 1 h (*p* < 0.01) (Figure 3c,d). In contrast, no group difference was observed in the mRNA levels of *Dnmt1* (*F*_1, 103, retrieval_ = 2.13, *p* = 0.15; F_5, 103, test_ = 1.12, *p* = 0.35; *F*_5, 103, interaction_ = 0.21, *p* = 0.96; Figure 3b). Together, these results indicate that DNMT3a and/or DNMT3b, but not DNMT1, mediates DNA methylation during memory reconsolidation, and that DNA methylation may occur at 1 h after memory retrieval.

### 3.3. Gnas Is the Key Gene in the BLA That Is Methylated during Reconsolidation of Morphine-related Memories

DNA methylation was analysed 1 h after memory retrieval. A total of 62 DMRs were identified (difference in DNA methylation level > 0.2, *p* < 0.05, CpG number ≥ 5), which included 43 hypermethylated DMRs and 19 hypomethylated DMRs. Because the effects of DMRs located in gene bodies on the expression of genes are not as clear as those of DMRs located in gene promoter regions, we focused on genes with DMRs located in their promoter regions. Thirteen DMRs were found located in the promoter regions of genes, including nine hypermethylated DMRs and four hypomethylated DMRs (Table 2). Using bisulphate PCR, we further validated the DMRs located in the promoter regions of *Gnas*, *Sox10*, and *Pirk3r1*, which are genes involved in neural plasticity and memory modulation, and found that only the DMR located in the *Gnas* promoter could be validated (*p* < 0.0001, Student’s *t*-test, Figure 4b, two samples, each consisting of the pooled tissues of six rats).

In order to investigate whether upregulation of the DNA methylation level of the *Gnas* promoter affected the gene expression level of *Gnas*, we performed quantitative reverse transcription polymerase chain reaction (RT-qPCR) and found that *Gnas* gene expression was reduced at 1 h after memory retrieval (*n* = 4, *t* = 3.80, *p* < 0.01, Figure 4c), as compared to the no-retrieval group (*n* = 5).

## 4. Discussion

Our study indicated a role for region-specific DNA methylation in the reconsolidation of morphine-related memories in rats. We found that micro-injection of 5-aza into the BLA, but not into the CeA or NAc, impeded the reconsolidation of mCPP. Our results also showed that the expression of *Dnmt3a* and *Dnmt3b*, but not *Dnmt1*, was upregulated during the reconsolidation. Specifically, *Gnas* may be one of the hypermethylated genes crucial for the reconsolidation of mCPP.

Accumulated evidence has suggested that DNA methylation plays an essential role in cocaine-induced behavioral sensitization [29], incubation of craving [19], and heritable cocaine-seeking motivation [30], and contributes to long-term fear memory and drug addiction-related memory [24,31,32]. Our previous studies have revealed that inhibition of Dnmts in the hippocampal CA1 blocked the acquisition and consolidation of morphine/cocaine-related reward memory and that DNA methylation in the prelimbic cortex was involved in the retrieval [21,22,33]. One hallmark of long-term memory storage is the labile state following retrieval, which suggests that a period of reconsolidation occurs. As in fear memory studies [32], we then found that reconsolidation of morphine withdrawal memory was impaired by injection of a Dnmts inhibitor into the agranular insular cortex or BLA [24]. In this study, we discovered that injection of 5-aza, an inhibitor of Dnmts, into the BLA, but not into the CeA, disrupted the reconsolidation of mCPP. Another study has also shown that Dnmts activity in BLA was required for the reconsolidation of cocaine-related memory in a self-administration model [34]. All these results suggest that DNA methylation in the BLA is crucial for the reconsolidation of drug-related memories, regardless of whether they have reward or aversive properties.

Moreover, our results showed that microinjection of 5-aza into the NAc did not affect the reconsolidation of mCPP. Reconsolidation of morphine or cocaine reward memory required the activity of the NMDA receptor, transcription factor Zif268, and protein synthesis in both the BLA [1,7,15,35] and the NAc [7,26], as well as extracellular signal-regulated kinase (ERK) activation in the BLA, but not in the NAc core [15]. Thus, the role of Dnmts in memory reconsolidation seems to be region-specific, similar to ERK activation. Another possibility is that, although Dnmts in the NAc also play a role in reconsolidation, DNA methylation in the NAc may occur at a different time window from that in the BLA; we did not investigate this possibility in the current study.

To find the appropriate time-point during memory reconsolidation at which to assess genes, we firstly evaluated the mRNA level of three subtypes of Dnmts (*Dnmt1*, *Dnmt3a*, and *Dnmt3b*) at different time-points after memory retrieval. Previous studies had shown that *Dnmt1* and *Dnmt3a/3b* prefer to methylate hemimethylated and unmethylated DNA, respectively [36]. We found that expression of *Dnmt3a/3b*, but not *Dnmt1*, was significantly upregulated, which suggested that de novo DNA methylation occurred during memory reconsolidation. Furthermore, *Dnmt3a* and *Dnmt3b* expression was significantly upregulated by 1 h after memory retrieval, and consistent with this finding, we found that injection of 5-aza at the same time-point had more powerful inhibitory effects on the reconsolidation than that immediately after memory retrieval. Moreover, the change in *Dnmt3b* expression might last up to 6 h, while the shift in *Dnmt3a* expression may last up to 12 h; Dnmts inhibition at 12 h after memory retrieval did not affect reconsolidation, and some studies have suggested that the period of reconsolidation lasts for up to 6 h after memory retrieval [5,37]. *Dnmt3b* may, therefore, play a more critical role in the reconsolidation than *Dnmt3a*, but this requires further investigation.

It is worth noting that the role of DNMT3a or DNMT3b in the regulation of addictive behavior or drug memory is still controversial. Laplant et al. (2010) found an up-regulation of Dnmt3a gene expression at 4 h and a down-regulation at 24 h after repeated cocaine administration, while DNMT3b expression did not change significantly [16]. However, Pol Bodetto S et al. (2013) reported that the DNMT3a and DNMT3b expression in rat CPu and PFCx were significantly upregulated at different time points after chronic cocaine treatment [38].

As epigenetic enzymes, Dnmts affect many genes, which begs the questions as to the identity of the target genes and whether DNA methylation exerts its role in reconsolidation via these targets. In the present study, we used RRBS to identify the genes that were methylated at 1 h after retrieval of mCPP. Our study indicated that DNA methylation is a highly dynamic process since 62 differentially methylated regions were identified in the BLA in response to memory reconsolidation (Table 3). Of the identified DMRs, 73.5% were located in intergenic regions. To date, most of the studies related to gene regulation by methylation have focused on gene promoters. It is, therefore, somewhat unusual that we found only 21.0% of DMRs located in promoter regions. However, it is worth noting that RRBS analyzed only a portion of genomic regions. Therefore, to analyze DNA methylation comprehensively, whole-genome bisulphite sequencing is required.

We further investigated the potential effect of DNA methylation on gene expression in the BLA. Because DNA hypermethylation in the promoter could inhibit gene expression [39], we analyzed the 14 genes with DMRs located in the promoter region (Table 2). Previous studies have shown that *PP1*, *fosB*, *DAT*, and *TACR3* were DNA methylated in the context of drug abuse [22,29,40,41]. However, most of the genes found in the present study have not been reported in this context previously. It is possible that these genes were reconsolidation-specifically methylated or that the RRBS procedure may not cover the whole genome. We focused on the ten hypermethylated genes (Table 2) since *Dnmt3a/3b* expression was upregulated during memory reconsolidation (Figure 3). Among these ten genes, *Gnas*, *Pik3r1*, and *Sox10* attracted our attention, because *Gnas* encodes the stimulatory G-protein subunit (Gαs) [42], *Pik3r1* is involved in the PI3K-Akt signaling pathway, and *Sox10* regulates transcription factor binding [43,44], all of which are processes crucial to learning and memory [45,46]. Hence, we specifically investigated whether *Gnas*, *Sox10*, and *Pik3r1* were target genes of Dnmts during reconsolidation. Using bisulphate PCR, we found that only methylation of the DMR located in the *Gnas* promoter could be validated (Figure 4 and Appendix A). As expected, the mRNA level of *Gnas* was downregulated. Taken together, these findings indicate that DNA methylation of the *Gnas* promoter may play a crucial role in the reconsolidation of morphine reward memory.

*Gnas* is known to be imprinted; its epigenetic alterations have been connected with neurological diseases, such as autism spectrum disorder and Parkinson’s disease [47,48,49]. Chronically increased Gαs signaling disrupts spatial learning [50]. Gαs-coupled dopamine D1 dopamine receptors or other G-protein-coupled receptors elicit activation of cyclic AMP/Protein kinase A [51]. Each of these signaling events has been implicated in neuroplasticity, drug addiction, and memory reconsolidation [51,52,53]. Further investigations are necessary to elucidate the pathway through which *Gnas* methylation affects reconsolidation.

There are several limitations of this study. First, we only examined the mRNA levels of Dnmts. Neither the protein level of Dnmts nor the effects of knocking down any subtype of Dnmts on the reconsolidation were detected. Therefore, which subtype of Dnmts mediates the DNA methylation during reconsolidation remains unknown. Second, we have not injected 5-aza into the CeA or the NAc one hour after retrieval to more convincingly rule out the role of Dnmts in reconsolidation in these brain regions.

Third, although we did not observe significant upregulation of Dnmts right after memory retrieval, injecting 2 µg 5-aza into the BLA immediately after retrieval reduced the mCPP score at RT 14, but not at RT1 or RT7 (Figure 1b). These data imply that 5-aza caused the instability of morphine reward memory, which could not be maintained 14 days after microinjection of 5-aza. This effect may be due to the little increase trend of *Dnmt3a* level even right after retrieval (Figure 3c), suggesting that DNA methylation occurs within 30 min after stimulation, as we previously hypothesized [22]. Microinjection of 5-aza into the BLA one hour after retrieval, on the other hand, interrupted the reconsolidation, resulting in memory failure 24 h later. DNA methylation in the BLA occurs incrementally during the reconsolidation process, which might explain the distinctive effects. The impact of 5-aza varies depending on the injection time points. According to our findings, DNA methylation in the BLA may be increased immediately following memory recall and reach its highest point one hour later. As a result, one explanation for our findings is that 5-aza took action shortly after memory retrieval but exhibited a weaker inhibitory impact than when administered one hour later. When DNA methylation reaches its highest point, most of the 5-aza in the BLA may be metabolized, resulting in a lower dose of 5-aza and a weaker effect on reconsolidation. Similar results also showed that relatively small amounts of protein synthesis inhibitors had weak effects on reconsolidation or had no effect on this process [54]. More research is needed to confirm the above hypothesis further.

## 5. Conclusions

In summary, the present study implicated dissociable DNA methylation mechanisms in the BLA and NAc in the reconsolidation of opiate-related memory. Moreover, our results suggest that *Gnas* may be a novel target gene that plays a crucial role in this memory reconsolidation.

## Figures and Tables

**Figure 1 genes-13-00553-f001:**
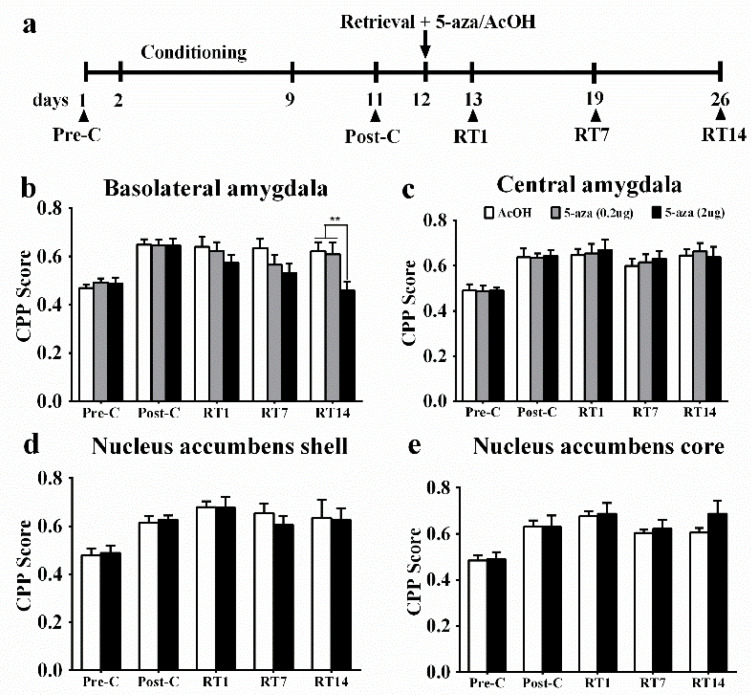
Effect of injection of 5-aza into the basolateral amygdala (BLA) and nucleus accumbens (NAc) on the reconsolidation of morphine reward memory. (**a**) Timeline of the experiment. (**b**) Injection of 5-aza (2 µg) into the BLA immediately after memory retrieval impaired memory reconsolidation of mCPP (*n* = 8–10). (**c**) Injection of 5-aza (0.2 µg or 2 µg) into the central amygdala (CeA) did not affect reconsolidation (*n* = 9–10). (**d**) Injection of 5-aza (2 µg) into the NAc shell did not affect reconsolidation (*n* = 7). (**e**) Injection of 5-aza (2 µg) into the NAc core did not affect reconsolidation (*n* = 7–11). Data are the mean ± SEM of preference scores during the CPP tests. ** *p* < 0.01, compared to acetic acid (AcOH) injection at the same time-point. The schematic representation of the injection sites: see Appendix A.

**Figure 2 genes-13-00553-f002:**
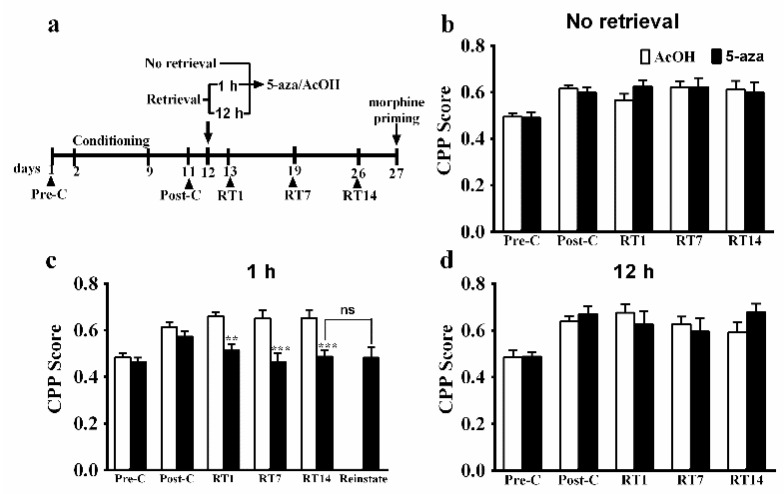
The effect of 5-aza on morphine reward memory reconsolidation was retrieval- and time-window-dependent. (**a**) Timeline of the experiment. (**b**) Infusion of 5-aza into the basolateral amygdala (BLA; 2 µg) had no effect on reconsolidation in the absence of retrieval (*n* = 13). (**c**) Intra-BLA infusion of 5-aza (2 µg) 1 h after memory retrieval blocked reconsolidation (*n* = 9–12). (**d**) Injection of 5-aza into the BLA at 12 h after retrieval did not affect reconsolidation (*n* = 7–8). Data are expressed as the mean ± SEM of preference scores. ** *p* < 0.01, *** *p* < 0.001, compared to AcOH at the same time-point. ns: not significant.

**Figure 3 genes-13-00553-f003:**
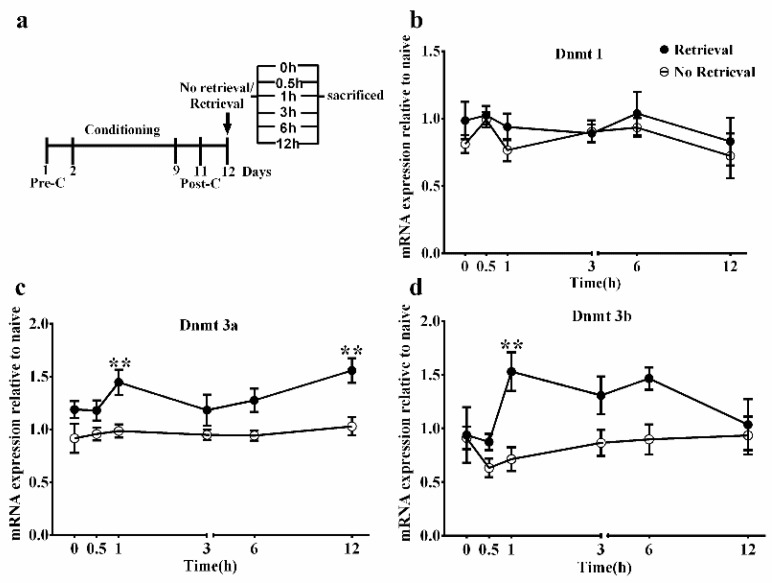
Transcription regulation of DNA methyltransferase genes (Dnmts) by morphine-associated memory reconsolidation in the basolateral amygdala (BLA). (**a**) Timeline of the experimental procedure. (**b**–**d**) Reverse transcription PCR analysis of *Dnmt1* (**b**, *n* = 9–10), *Dnmt3a* (**c**, *n* = 9–10), and *Dnmt3b* (**d**, *n* = 8–10) mRNA expression in the BLA during reconsolidation. Data are expressed as the mean ± SEM. ** *p* < 0.01, compared to no-retrieval group at the same time-point.

**Figure 4 genes-13-00553-f004:**
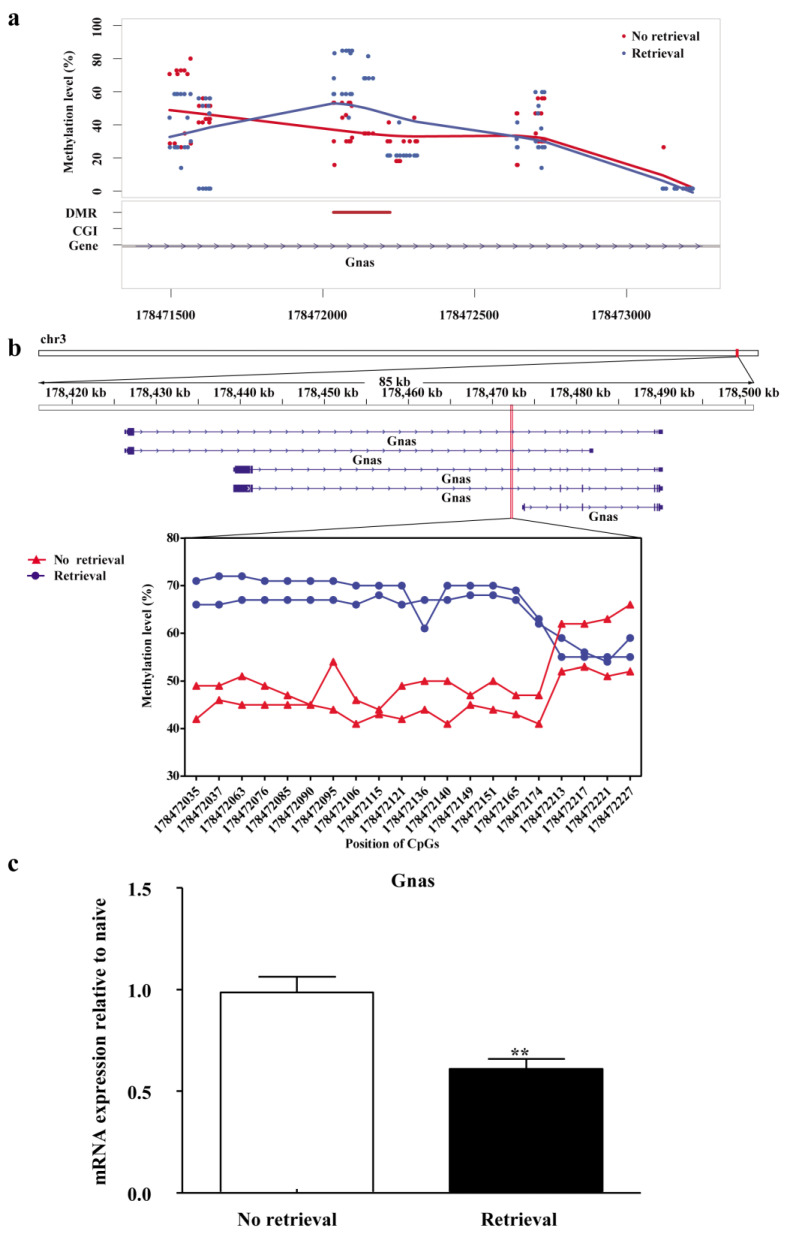
Correlation between DNA methylation and downregulation of mRNA expression at one hour after memory retrieval. (**a**) The DNA methylation level was evaluated by reduced-representation bisulphite sequencing. (**b**) The DNA methylation level was evaluated by bisulphite PCR validation. (**c**) Gene expression level of *Gnas* after memory retrieval (*n* = 4–5). ** *p* < 0.01, compared to no-retrieval group. The DNA methylation level of *Sox10* and *Pik3r1*: see Appendix A.

**Table 1 genes-13-00553-t001:** Primer sequences for RT-PCR.

Gene	Primer Sequence	
	Forward Sequence	Reverse Sequence
*Dnmt1*	TATTGCAGTCGCGGTCACTT	CTGATTGATTGGCCCCAGGT
*Dnmt3a*	TCTTGCTCACAAAGACCACGA	TACCACGGTTCTCCTCCTGT
*Dnmt3b*	ACCAGGCCTTGAAAACCTCAG	CATGGTTTCCTGCAAGTCCCT
*GAPDH*	ACCTTTGATGCTGGGGCTGGC	GGGCTGAGTTGGGATGGGGACT
*Gnas*	GGAGGACAACCAGACCAACC	GACCTTCTCAGCAAGCAGATC

**Table 2 genes-13-00553-t002:** Genes with differentially methylated regions (DMRs) in their promoter region.

Genes	Differentially Methylated Regions	Hypermethylated or Hypomethylated	*p* Value
*Zrsr1*	Chr14:107824490-107824759	Hypermethylated	7.57 × 10^−6^
*Gnas*	chr3:178472035-1784722218	Hypermethylated	2.43 × 10^−5^
*Pced1b*	Chr7:138329413-138329534	Hypermethylated	0.003298
*Amigo2*	Chr7:138329413-138329534	Hypermethylated	0.003298
*Sox10*	chr7:120397931-120398118	Hypermethylated	0.0102
*Pik3r1*	chr2:50965442-50965530	Hypermethylated	0.0107
*Tnnc2*	Chr3:167428575-167428739	Hypermethylated	0.01381041
*Cxcr5*	Chr8:47457610-47457753	Hypermethylated	0.024671
*Fkbpl*	chr20:6490315-6490406	Hypermethylated	0.0248
*RGD1563015*	chr5:142077035-142077175	Hypermethylated	0.0258
*Rps10*	chr20:9422414-9422470	Hypomethylated	1.41 × 10^−9^
*RGD1560608*	chr6:144559138-144559194	Hypomethylated	0.0019
*Crb2*	chr3:27274071-27274146	Hypomethylated	0.0224
*Serpinh1*	chr1:170510550-170510601	Hypomethylated	0.0252

**Table 3 genes-13-00553-t003:** The distribution of DMRs.

Gene Name	Position of DMR	CpG Number of the DMR	*p* Value	Promoter	Gene Body	Upstream	Downstream
*Aff1*	chr14:7218380-7218515	9	0.00207539		Y		
*Amigo2*	chr7:138329413-138329534	10	0.003297764	Y	Y		
*Ano4*	chr7:29919053-29919161	10	0.01020776		Y		
*Arhgap31*	chr11:67463804-67463994	14	0.000473217		Y		
*Arhgef2*	chr2:207404742-207404827	6	0.009214125		Y		
*Bcl11b*	chr6:141020775-141021154	15	8.95 × 10^−10^		Y		
*Bcl11b*	chr6:141052868-141053020	6	0.02759777		Y		
*Bmp7*	chr3:176986680-176986773	5	0.003185848		Y		
*Btbd17*	chr10:102994047-102994225	8	0.02664855		Y		
*C4a*	chr20:6490315-6490406	6	0.02475037		Y		
*C4a*	chr20:6530195-6530367	11	0.006960988		Y		
*Cacna2d2*	chr8:115596160-115596246	6	0.003831569		Y		
*Capn2*	chr13:105860689-105860886	7	0.000954839		Y		
*Clstn1*	chr5:170207211-170207370	8	0.009061262		Y		
*Col16a1*	chr5:152027022-152027215	6	0.001871846		Y		
*Commd1*	chr14:107824490-107824759	27	7.57 × 10^−6^		Y		
*Crb2*	chr3:27274071-27274146	5	0.02240409	Y		Y	
*Cxcr5*	chr8:47457610-47457753	8	0.02467058	Y		Y	
*Disc1*	chr19:68763258-68763457	18	0.000567238		Y		
*Dpysl2*	chr15:48089025-48089178	14	3.45 × 10^−13^		Y		
*Dscaml1*	chr8:48478120-48478263	7	0.000375843		Y		
*Eif4b*	chr7:141499843-141500003	8	0.0045236				Y
*Fkbpl*	chr20:6490315-6490406	6	0.02475037	Y		Y	
*Gipc2*	chr2:275973554-275973698	7	0.000207575				Y
*Gnas*	chr3:178441267-178441419	19	1.28 × 10^−12^		Y		
*Gnas*	chr3:178472035-178472221	22	2.43 × 10^−5^	Y	Y	Y	
*Hm13*	chr3:154572338-154572523	17	1.25 × 10^−6^		Y		
*Hs3st3b1*	chr10:50167234-50167300	8	0.001685876		Y		
*Jak3*	chr16:19972561-19972636	5	0.01845945		Y		
*Jup*	chr10:88094186-88094336	7	0.006709436		Y		
*Kazn*	chr5:164412676-164412854	12	0.004983534		Y		
*Kcnh3*	chrX:115381900-115382098	7	0.000139526		Y		
*Kif19*	chr10:102994047-102994225		0.02664855				Y
*Kif26b*	chr13:101990776-101990963	5	0.02067161		Y		
*Lrrk1*	chr1:128250485-128250670	10	0.001559071		Y		
*Lrrn2*	chr13:54749407-54749591	5	0.01956404		Y		
*Mcoln2*	chr2:270577752-270577951	12	0.000800436		Y		
*Mecom*	chr2:137270802-137270931	5	0.0126658		Y		
*Mtss1*	chr7:99426687-99426843	6	0.01163478		Y		
*Mvd*	chr19:65970858-65970974	6	0.00726908		Y		
*Narfl*	chr10:14960107-14960271	16	7.19 × 10^−6^		Y		
*Nfix*	chr19:36874870-36874950	11	7.19 × 10^−5^		Y		
*Oxct1*	chr2:72940605-72940773	5	0.0135552		Y		
*Pced1b*	chr7:138329413-138329534	10	0.003297764	Y		Y	
*Per3*	chr5:171694018-171694214	12	0.003234661		Y		
*Pik3r1*	chr2:50965442-50965530	6	0.01074433	Y		Y	
*Prr5*	chr7:125324729-125324849	6	5.39 × 10^−6^		Y		
*Ptpn11*	chr12:42770644-42770806	10	0.000300016		Y		
*Ptprm*	chr9:114967504-114967678	8	0.009920264		Y		
*Pygm*	chr1:228746598-228746766	8	0.005512662		Y		
*RGD1560608*	chr6:144559138-144559194	7	0.001928227	Y	Y		
*RGD1563015*	chr5:142077035-142077175	6	0.02576382	Y	Y		
*Rhbdd1*	chr9:87975914-87975949	8	0.02561427				Y
*Rps10*	chr20:9422414-9422470	8	1.41 × 10^−9^	Y	Y		
*Sall1*	chr19:34396734-34396837	7	0.01097885				Y
*Sema5b*	chr11:71436403-71436588	7	0.01134854		Y		
*Serpinh1*	chr1:170510550-170510601	6	0.02524889	Y	Y		
*Sfxn5*	chr4:181583937-181584095	10	0.000178655		Y		
*Sox10*	chr7:120397931-120398118	8	0.01023295	Y		Y	
*Spats2*	chrX:115381900-115382098	7	0.000139526		Y		
*Stk19*	chr20:6490315-6490406	6	0.02475037		Y		
*Stk19*	chr20:6530195-6530367	11	0.006960988		Y		
*Tmem63a*	chr13:104248401-104248590	10	0.01664342		Y		
*Tnnc2*	chr3:167428575-167428739	15	0.01381041	Y		Y	
*Tspan5*	chr2:262646069-262646263	8	0.002377802		Y		
*Txndc15*	chr17:11251263-11251421	18	1.58 × 10^−6^		Y		
*Wnt7b*	chr7:126158090-126158176	8	0.002492002		Y		
*Zfp503*	chr15:2361950-2362100	15	8.09 × 10^−5^				Y
*Zrsr1*	chr14:107824490-107824759	27	7.57 × 10^−6^	Y		Y	

## Data Availability

The data reported in this study are available from the corresponding author upon reasonable request.

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
