# Peer review of "Gnas Promoter Hypermethylation in the Basolateral Amygdala Regulates Reconsolidation of Morphine Reward Memory in Rats"

_genes, 2022, doi:10.3390/genes13030553_

Round 1
Reviewer 1 Report
Summary
This paper from Liu and collogues describes a study about the relationship between DNA methylation and memory consolidation using Rat as their research model. The authors first tested reconsolidation of morphine induced reward memory in rats that were injected with DNMT inhibitor and revealed that inhibition of DNMTs in BLA but not in NAc impairs memory re -consolidation. The authors further found that Dnmt3a and b were upregulated in BLA after memory retrieval using Q-PCR, suggesting DNMTs may involve in memory reconsolidation. Finally, they performed RRBS ad bisulphite PCR to identify downstream target genes of DNMT3a/b and found several hits that change their methylation status. In particular, they focused on Gnas and showed downregulation of Gnas in BLA after memory retrieval, suggesting Gnas may play a critical role in memory consolidation.
Overall, the research topic of this paper is important, which is also fit for the scope of the journal. The experiments performed in this paper is well designed and the data is clearly presented. The paper is well-structured and clear written. I think this is an interesting paper and only have a few comments to improve it.
Major comments
- In experiment 1, additional information is needed for supporting the specificity of the pharmacological experiments. Are DNMTs the only targets of 5-aza? Will 5-aza cause any anatomical changes in the injected brain regions?
- The authors found that Dnmt3a and b were upregulated during memory retrieval at the transcriptional level, what about the protein level? This should not be a difficult experiment as the antibodies are available. Also testing animals carrying Dnmt3 mutations in the same behavioral assay would be helpful to further support DMNTs is functionally important during memory consolidation.
- Western blot for testing the expression of Gnas was described in the experiments procedures (line 156-170), but I couldn’t find the related results.
Minor comments
- b: in the figure, the grey and black bar labelled with 0.2ug and 2ug of 5-aza respectively, which is inconsistent with the figure legends (1 ug/side), the same problems occur to panel c-e.
- Figure2: what is the concentration of 5-aza used here, why the concentration was different compared to Figure1 if the legends are correct.
- Table3 needs to be remade as the current version is confusing. Adding lines to separate each raw will be helpful.
- Line 65:what does “CeA” stand for?
- Line 304: replace “levels” to “level”, “were” to “was”.
- Line 417: replace “DNMT3a” to “Dnmt3a”.
Author Response
Dear Reviewer,
Thanks a lot for the opportunity to revise our manuscript.
We appreciate the reviewer for his/her valuable comments, which are very helpful for improving our manuscript. We have studied the comments carefully and have made corrections. We hope the corrections can meet with approval. Revisions are marked in blue in the manuscript.
We are looking forward to hearing from you soon.
We have addressed the comments and revised the manuscript as follows:
Editor and Reviewer comments:
This paper from Liu and collogues describes a study about the relationship between DNA methylation and memory consolidation using Rat as their research model. The authors first tested reconsolidation of morphine induced reward memory in rats that were injected with DNMT inhibitor and revealed that inhibition of DNMTs in BLA but not in NAc impairs memory re -consolidation. The authors further found that Dnmt3a and b were upregulated in BLA after memory retrieval using Q-PCR, suggesting DNMTs may involve in memory reconsolidation. Finally, they performed RRBS ad bisulphite PCR to identify downstream target genes of DNMT3a/b and found several hits that change their methylation status. In particular, they focused on Gnas and showed downregulation of Gnas in BLA after memory retrieval, suggesting Gnas may play a critical role in memory consolidation.
Overall, the research topic of this paper is important, which is also fit for the scope of the journal. The experiments performed in this paper is well designed and the data is clearly presented. The paper is well-structured and clear written. I think this is an interesting paper and only have a few comments to improve it.
Response: We thank the reviewer very much for the constructive comments. We have studied the comments carefully and have made corrections accordingly.
Major comments
In experiment 1, additional information is needed for supporting the specificity of the pharmacological experiments. Are DNMTs the only targets of 5-aza? Will 5-aza cause any anatomical changes in the injected brain regions?
Response: We thank the reviewer for these comments. DNMTs are the main targets of 5-aza. Previously, we injected 5-aza into the BLA and did not find it affected the locomotor activities or the preference/aversion of drug-naive rats (Liu, Zhang, Li, & Sui, 2016; Zhang, Han, & Sui, 2014). We have added this information in experiment 1. Please see lines 205-207.
The authors found that Dnmt3a and b were upregulated during memory retrieval at the transcriptional level, what about the protein level? This should not be a difficult experiment as the antibodies are available. Also testing animals carrying Dnmt3 mutations in the same behavioral assay would be helpful to further support DMNTs is functionally important during memory consolidation.
Response: We agree with the reviewer that it is better to confirm the effects of DNMT3 on the reconsolidation via testing the protein level and testing animals carrying Dnmt3 mutations. The change of mRNA and protein level usually showed a strong and significant positive correlation. To use as few rats as possible, we have not tested the protein level. We have not used DNMT3 mutated rats because it is difficult for us to edit genes in rats. We have discussed this limitation of this study. In the future, we are planning to conduct the suggested experiments if the mutated rats are ready.
Western blot for testing the expression of Gnas was described in the experiments procedures (line 156-170), but I couldn’t find the related results.
Response: We are sorry for this mistake. We have deleted this part of experiments procedure.
Minor comments
b: in the figure, the grey and black bar labelled with 0.2ug and 2ug of 5-aza respectively, which is inconsistent with the figure legends (1 ug/side), the same problems occur to panel c-e.
Response: We appreciate this comment. If each rat received two sides injection (1 ug/side), as a result, each rat got 2ug of 5-aza. We have replaced 1 ug/side with 2 ug in the figure legends to make it clear.
Figure2: what is the concentration of 5-aza used here, why the concentration was different compared to Figure1 if the legends are correct.
Response: Thank you very much for the comment. The concentration of 5-aza was 2 µg/µl, and we also used 2 ug 5-aza in figure 2. We have added this information in the figure legends.
Table3 needs to be remade as the current version is confusing. Adding lines to separate each raw will be helpful.
Response: We appreciate this comment and agree with the reviewer. Because the journal has requirements for the format of tables, we have reduced the font size in Table 3 instead. We hope this may be clear. Please see pages 11-14.
Line 65:what does “CeA” stand for?
Response: This is a helpful comment. Thank you very much. CeA stands for the central amygdala, and we have added this full name while mentioning it for the first time. Please see line 67.
Line 304: replace “levels” to “level”, “were” to “was”.
Response: We appreciate this comment, and we have replaced these words. Please see line 297.
Line 417: replace “DNMT3a” to “Dnmt3a”.
Response: This is a helpful comment, and we have replaced this word. Please see line 415.
Reviewer 2 Report
The manuscript by Liu et al., studies the role of methylation in the process of reconsolidation of morphine reward memory. The authors demonstrated that inhibiting DNA methylation by injecting 5-aza into the basolateral amygdala 10 minutes after memory retrieval causes slight impairment in reconsolidation of morphine reward memory. They further demonstrated that the effects of 5-aza are more pronounced when it is injected 1 hour after retrieval rather than 12 hours after retrieval. Using a combination of several molecular biology techniques they reported that two different DNA methylation genes are upregulated in BLA right after memory retrieval and also identifies several candidate genes whose promoters are differentially regulated in BLA following memory retrieval. They tested genes which has previously been demonstrated to be involved in memory modulation and verified that the promoter region of Gnas gene is indeed hypermethylated and the gene downregulated following memory retrieval. The paper present interesting findings, the experiments are well designed and the interpretation is backed by rigorous statistical tests. However, there are some points that the authors need to address:
- Injection of 5-aza after 1 hour in Figure 2C causes severe impairment of memory reconsolidation which matches nicely with transcriptional regulation data of Dnmts. However, although no significant upregulation of Dnmts was observed right after memory retrieval (i.e. atleast upto 30 minutes), injection of 5-aza after just 10 minutes of memory retrieval caused decrease in memory reconsolidation in Figure 1B which seemed counter-intuitive. The authors need to explain this in the text. Also, the authors need to provide some explanations regarding why in Figure 1B the effects are visible only after RT14.
- Although a major effect of injection of 5-aza into BLA was observed after 1 hour following memory retrieval, for the other regions the injection was performed only at one time point viz. 10 minutes after memory retrieval. The authors needed to inject 5-aza into the other regions (CA, NAA) I hour following memory retrieval to more convincingly rule out a role of Dnmts in memory reconsolidation in these brain regions.
- The authors need to perform 5-aza injections at more time points between 1 hour and 12 hours following memory retrieval to demonstrate till what time following memory retrieval differential DNA methylation is important in memory reconsolidation.
- It was not clear why the authors used 0.8% acetic acid as a control for 5-aza.
- For some of the abbreviated or shortened forms (for eg. 5-aza, DMR) the authors need to mention the full name while mentioning them for the first time at least.
- Please correct minor grammatical mistake in line 49, Page 2 (‘removal of’ instead of ‘removed’)
- The fonts of the axis titles and legends within the figures are too small making them very difficult to read. Please make them larger.
- Although the subfigures are denoted with small letters in the figures themselves, they are referred to with capital letters in figure legends and also in the text. Please make sure they are consistent.
- Please provide details (company, catalogue number) for all chemicals and reagents used.
Author Response
Dear Reviewer,
Thanks a lot for the opportunity to revise our manuscript.
We appreciate the reviewer for his/her valuable comments, which are very helpful for improving our manuscript. We have studied the comments carefully and have made corrections. We hope the corrections can meet with approval. Revisions are marked in blue in the manuscript.
We are looking forward to hearing from you soon.
We have addressed the comments and revised the manuscript as follows:
Editor and Reviewer comments:
The manuscript by Liu et al., studies the role of methylation in the process of reconsolidation of morphine reward memory. The authors demonstrated that inhibiting DNA methylation by injecting 5-aza into the basolateral amygdala 10 minutes after memory retrieval causes slight impairment in reconsolidation of morphine reward memory. They further demonstrated that the effects of 5-aza are more pronounced when it is injected 1 hour after retrieval rather than 12 hours after retrieval. Using a combination of several molecular biology techniques they reported that two different DNA methylation genes are upregulated in BLA right after memory retrieval and also identifies several candidate genes whose promoters are differentially regulated in BLA following memory retrieval. They tested genes which has previously been demonstrated to be involved in memory modulation and verified that the promoter region of Gnas gene is indeed hypermethylated and the gene downregulated following memory retrieval. The paper present interesting findings, the experiments are well designed and the interpretation is backed by rigorous statistical tests. However, there are some points that the authors need to address:
Response: We thank the reviewer very much for the constructive comments and the patient revision of our paper. We have studied the comments carefully and have made corrections accordingly.
Injection of 5-aza after 1 hour in Figure 2C causes severe impairment of memory reconsolidation which matches nicely with transcriptional regulation data of Dnmts. However, although no significant upregulation of Dnmts was observed right after memory retrieval (i.e. at least upto 30 minutes), injection of 5-aza after just 10 minutes of memory retrieval caused decrease in memory reconsolidation in Figure 1B which seemed counter-intuitive. The authors need to explain this in the text. Also, the authors need to provide some explanations regarding why in Figure 1B the effects are visible only after RT14.
Response: We appreciate this comment and agree with the reviewer. We have added one paragraph to discussion these findings.
Third, although we did not observe significant upregulation of Dnmts right after memory retrieval, injecting 2 µg 5-aza into the BLA immediately after retrieval reduced the mCPP score at RT 14, but not at RT1 or RT7 (Figure 1b). This data implies that 5-aza caused the instability of morphine reward memory, which could not maintain 14 days after microinjection of 5-aza. This effect may be due to the little increase trend of Dnmt3a level even right after retrieval (Figure 3c), suggesting that DNA methylation occurs whin 30 min after stimulation, as we previously hypothesized(Zhang et al., 2014). Microinjection of 5-aza into the BLA one hour after retrieval, on the other hand, interrupted the reconsolidation, resulting in memory failure 24 hours later. DNA methylation in the BLA happens incrementally during the reconsolidation process, which might explain the distinctive effects. The impact of 5-aza varies depending on the injection time points. According to our findings, DNA methylation in the BLA may be increased immediately following memory recall and reach its highest point one hour later. As a result, one explanation for our findings is that 5-aza took action shortly after memory retrieval but had a weaker inhibitory impact than when administered one hour later. When DNA methylation reaches its highest point, most of the 5-aza in the BLA may be metabolized, resulting in a lower dose of 5-aza and a weaker effect on reconsolidation. Similar results also showed that relatively small amounts of protein synthesis inhibitors weak reconsolidation or have no effect on this process (Yu, Chang, & Gean, 2013). More research is needed to confirm the above hypothesis further.
Please see lines 459-484.
Although a major effect of injection of 5-aza into BLA was observed after 1 hour following memory retrieval, for the other regions the injection was performed only at one time point viz. 10 minutes after memory retrieval. The authors needed to inject 5-aza into the other regions (CA, NAA) I hour following memory retrieval to more convincingly rule out a role of Dnmts in memory reconsolidation in these brain regions.
Response: We appreciate this comment. We have added this in the discussion as a limitation of this study. Please see lines 462-464.
The authors need to perform 5-aza injections at more time points between 1 hour and 12 hours following memory retrieval to demonstrate till what time following memory retrieval differential DNA methylation is important in memory reconsolidation.
Response: We appreciate this comment. According to the protocols which were approved by the Research Ethics Review Board of Institute of Psychology, Chinese Academy of Sciences, we should try our best to reduce the number of used rats. We only tested the time points, which are necessary to confirm the reconsolidation (two time points within the reconsolidation window, one time point outside the window, and no retrieval).
It was not clear why the authors used 0.8% acetic acid as a control for 5-aza.
Response: We appreciate this comment. Because 5-aza was dissolved in 0.8% acetic acid, when we injected 5-aza into the brain, 0.8% acetic acid was also been injected. To exclude the potential effect of the vehicle, we used 0.8% acetic acid as a control.
For some of the abbreviated or shortened forms (for eg. 5-aza, DMR) the authors need to mention the full name while mentioning them for the first time at least.
Response: Thank you for this helpful comment. We have added the full name when mentioning them for the first time.
Please correct minor grammatical mistake in line 49, Page 2 (‘removal of’ instead of ‘removed’)
Response: We appreciate this comment. We have corrected it accordingly.
The fonts of the axis titles and legends within the figures are too small making them very difficult to read. Please make them larger.
Response: Thank you for this helpful comment. We have used larger fonts in the figures.
Although the subfigures are denoted with small letters in the figures themselves, they are referred to with capital letters in figure legends and also in the text. Please make sure they are consistent.
Response: We appreciate this comment. We have replaced all the capital letters with small letters.
Please provide details (company, catalogue number) for all chemicals and reagents used.
Response: Thank you for this helpful comment. We have added these details.